# END-TO-END UNIFIED DENSE 3D GEOMETRY AND MOTION PERCEPTION

## ABSTRACT

Predicting 3D geometry and motion from videos is crucial for various applications. Most existing methods adopt a two-stage reconstruct-then-tracking pipeline, which first perceives 3D geometry and then exploits this 3D information to track each pixel. They usually employ the conventional iterative tracking strategy and are thus inefficient, especially for dense motion estimation. Moreover, they fail to leverage the complementary motion information for better dynamic reconstruction. To address these limitations, we propose MotionVGGT, an end-to-end unified transformer architecture that simultaneously perceives dense 3D geometry, camera pose, and motion. We introduce a set of geometry, camera, and motion tokens to represent each frame and interact with each other through interleaved frame attention and global attention layers. We then employ multiple heads to decode point maps, camera poses, and 3D motions from the corresponding tokens. Specifically, we design a conditional dense prediction head and use the motion tokens as conditions to modulate the decoding process of geometry tokens to transform them into motions. Our model directly generates dense per-pixel 3D motion fields in a single forward pass without external trackers. By unifying geometry and motion modeling, MotionVGGT further equips visual geometry foundation models with motion awareness. Our MotionVGGT shows a strong generalization ability across diverse visual geometry perception tasks, establishing a practical and universal paradigm for more comprehensive scene understanding.

## 1 INTRODUCTION

The physical world is essentially a dynamic three-dimensional (3D) environment. Embodied agents observe the world through continuous video streams, where objects move, deform, interact, and occlude one another over time. For accurate and safe perception and planning, it is crucial to understand both 3D geometry and dynamic motion from videos, demanding an evolving dynamic scene representation that is coherent across frames (Zhu & Tang, 2025; Shao et al., 2023; Choe et al., 2023; Tretschk et al., 2024).

Recent years have witnessed rapid progress in geometry foundation models. 3D reconstruction methods (Wang et al., 2025b) predict a set of 3D attributes, including camera parameters and depth/point maps in a single forward pass, achieving competitive or state-of-the-art results across multiple 3D tasks. 2D Point tracking methods (Karaev et al., 2024) jointly track large sets of pixels in long videos, modeling inter-point dependencies to improve accuracy and robustness under occlusion and out-of-view re-entries. Building on top of them, 3D point tracking methods (Karaev et al., 2024; Xiao et al., 2024; Ngo et al., 2025) adopt a pipeline that proceeds from 3D reconstruction to tracking, thereby sequentially extracting motion information. However, their obtained tracks are usually sparse and are only anchored to the source frame, leading to incomplete motion modeling. Moreover, it fails to exploit the rich motion cues that could otherwise enhance dynamic scene reconstruction, thus facing drawbacks in performance and inefficiency.

Motivated by this, we introduce MotionVGGT, an end-to-end unified transformer architecture that simultaneously perceives dense 3D geometry and motion. It only takes as inputs RGB video or image sequences and predicts both geometry (camera, depth, point maps) and dense 3D motion fields in a single forward pass, as shown in Figure 1. We introduce a set of geometry, camera, and motion tokens to represent each frame, which interact through interleaved frame attention and

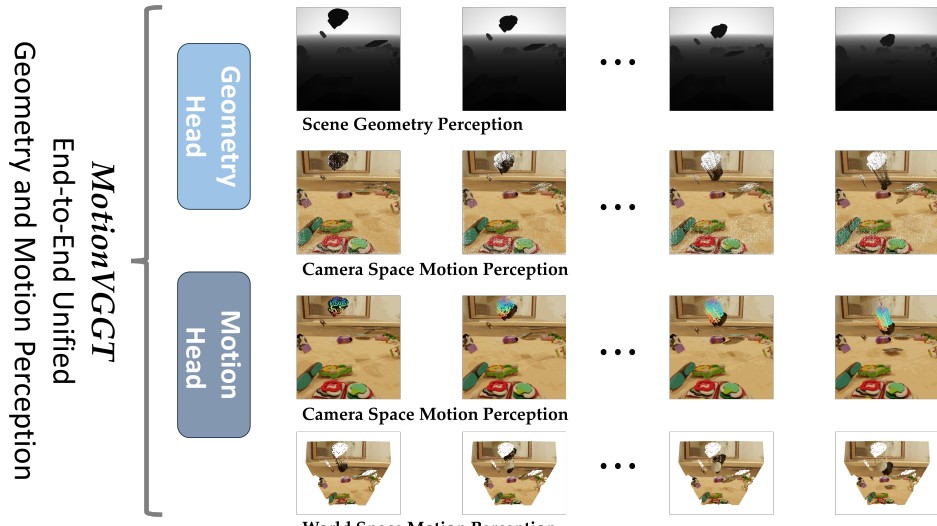

Figure 1: **Overview of our proposed MotionVGGT.** MotionVGGT achieves unified geometry and motion perception in world space in an-end-to-end manner.

global attention layers. Multiple decoding heads are then employed to recover point maps, camera poses, and 3D motions from the corresponding tokens. In particular, we design a conditional dense prediction head, where motion tokens serve as conditions to modulate the decoding of geometry tokens, effectively transforming them into motion representations. Our method requires no optical flow, no external tracker, and no extra geometric inputs, yielding image-wide dense 3D motion and aligning naturally with dynamic scene reconstruction. With MotionVGGT, we extend visual geometry foundation models from static geometry reconstruction to dynamic unified perception, offering a simple, scalable, and training-friendly path toward general scene understanding and broad deployment in autonomous driving, embodied AI, and AR/VR.

## 2    RELATED WORK

**Classical SfM/MVS.** Early methods reconstruct 3D geometry from unordered image collections by alternating correspondence discovery, camera pose estimation, and dense multi-view supervision. COLMAP (Schonberger & Frahm, 2016) remains a widely used, high-quality baseline with mature Structure-from-Motion (SfM) and Multi-View Stereo (MVS) modules. DeepMVS (Huang et al., 2018) has progressively shifted from hand-crafted matching to learned cost volumes and transformers, improving accuracy and scalability.

**Pairwise and multi-view geometric priors.** A growing family of methods reframes correspondence and reconstruction as direct point-map regression, reducing reliance on calibration and optimization and providing strong geometric priors for downstream tasks. DUSt3R (Wang et al., 2024) regresses dense point maps from uncalibrated and unposed image pairs and recovers accurate scene geometry with a single forward pass. MASt3R (Leroy et al., 2024a) augments this paradigm with dense local features and a 3D-grounded matching head, improving robustness under extreme viewpoint changes and accelerating matching. Recent works extend these ideas from pairs to multi-view settings. Fast3R (Yang et al., 2025) processes hundreds to thousands of unposed images jointly in a single forward pass for efficient multi-view reconstruction. Point3R (Wu et al., 2025) targets dense streaming reconstruction with an explicit spatial pointer memory that integrates incoming observations into a global coordinate frame.

**Visual geometry transformers.** VGGT (Wang et al., 2025b) predicts camera parameters, depth and point maps, and queried point tracks from one to hundreds of views in a single forward pass, and it often matches or exceeds methods that rely on heavy post-optimization. Recent variants improve scalability and latency. VGGT-Long (Deng et al., 2025) targets kilometer-scale sequences by chunking long videos and aligning overlapping segments with lightweight loop closure.

StreamVGGT (Zhuo et al., 2025) processes frames online with a causal transformer and cached memory for low-latency streaming reconstruction. FastVGGT (Shen et al., 2025) accelerates inference through a training free token-merging strategy and reports up to four times faster throughput on long sequences. However, most geometry-aware models are trained and evaluated on static scenes and do not natively output dense 3D motion fields over time, which motivates architectures that unify geometry with motion.

**Query-based 2D/2.5D tracking.** Point tracking studies motion by tracing queried points across long videos. TAP-Vid (Doersch et al., 2022) established a benchmark for this problem, emphasizing visibility changes and long horizons. TAPIR (Doersch et al., 2023) introduced a two-stage matcher-refiner that tracks arbitrary points efficiently and robustly. CoTracker (Karaev et al., 2024) demonstrated that jointly modeling inter-point dependencies markedly improves robustness to occlusions and out-of-view re-entries and enables simultaneous tracking of thousands of points. However, these methods follow selected points (often in 2D image space), not an image-wide dense 3D motion field. To achieve per-pixel coverage would require tracking vast numbers of queries, which increases computation and memory substantially.

**Dense 3D tracking.** DELTA (Ngo et al., 2025) proposes per-pixel 3D trajectories from monocular video with a coarse-to-fine design, reporting state-of-the-art accuracy and substantially higher throughput. Although DELTA outputs dense 3D tracks, it typically does not estimate explicit scene geometry jointly within the same model. Moreover, 3D tracking task is different with 3D motion. In the literature, "3D motion" often refers to per-point trajectories expressed in a world coordinate, representing how points move globally over time. In contrast, "3D tracking" is defined in each camera's per-frame coordinate system. Thus, even when DELTA recovers dense 3D tracks per frame, these tracks cannot be directly used for dynamic scene reconstruction in the world coordinate frame unless camera poses are provided.

**Dynamic Scene Reconstruction.** Dynamic scene reconstruction remains less explored compared with static scene reconstruction. Dynamic Point Map proposes a time invariant point map across timestamps. DynaDUSt3R (Jin et al., 2024) adds pixel level motion supervision to model dynamic geometry. Driv3R (Fei et al., 2024), applies a DUSt3R-based streaming 4D reconstruction framework to autonomous driving scenes. It regresses per-frame point maps, uses a temporal-spatial memory to reason dynamics, and aligns all frames into a consistent world coordinate system in an optimization-free manner. CUT3R (Wang et al., 2025d) maintains a persistent state and updates a global point map from streaming video for static and dynamic scenes. It predicts metric-scale point maps online and can even infer unseen views. More recently, D²USt3R has been proposed to extend DUSt3R into dynamic scenes by regressing 4D pointmaps that simultaneously capture static geometry and per-frame motion, thereby embedding spatio-temporal dense correspondences directly into the reconstruction process (Han et al., 2025).

## 3 METHOD

MotionVGGT is designed to capture and describe the 3D geometry and motion in dynamic scenes from videos or image sequences, a capability that prior geometry-aware models trained on static 3D scenes do not provide. To achieve this, we propose a novel motion head that disentangles 3D motion information from the geometry tokens produced by a pretrained geometry-aware backbone and predicts dense, per-pixel 3D motion fields. In this section, we first introduce the unified 3D geometry and motion perception task in Section 3.1. Subsequently, we describe our proposed architecture in Section 3.2. Finally, we provide the details of our training setup in Section 3.3.

### 3.1 UNIFIED GEOMETRY AND MOTION PERCEPTION

Most existing methods decompose dynamic scene understanding into two steps: first reconstructing geometry, and then applying tracking to model the motion of dynamic objects. This setup is inherently limited, as contemporary tracking methods generally rely on sparse query points or object instances, not dense matches, hindering fine-grained motion modeling. Moreover, conventional tracking is usually formulated in the camera coordinate system of each frame, which only provides per-frame trajectories. In contrast, 3D motion should be expressed in a consistent world coordinate

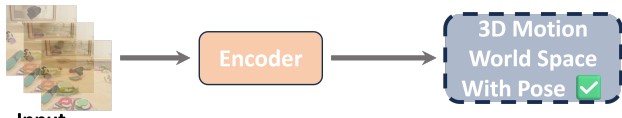

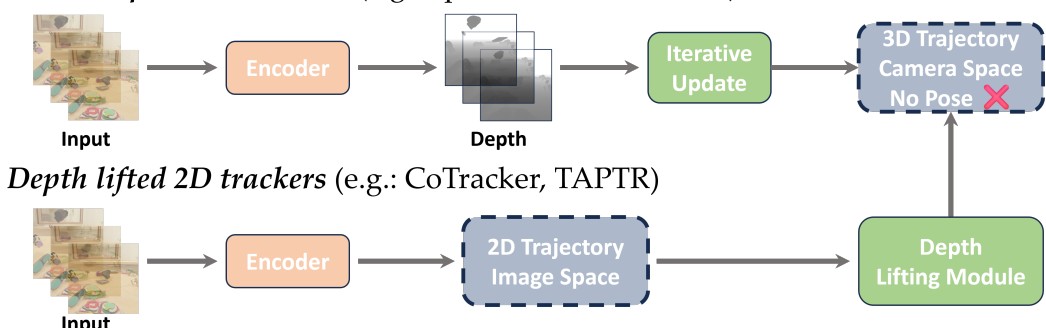

Figure 2: **Comparisons on 3D point tracking paradigms.** Our methodology directly obtains 3D motion information in world space, excelling in efficiency and simplicity.

system. The difference lies in the presence of camera pose:

$$\textbf{3D Motion:} \quad \mathcal{V}_{s \to t}(u, v) \in \mathbb{R}^3 \quad \text{in world coordinates,} \tag{1}$$

$$\textbf{3D Tracking:} \quad \mathcal{T} : \mathcal{O}_s \to \mathcal{O}_t \quad \text{in camera coordinates,} \tag{2}$$

where $\mathcal{V}_{s \to t}(u, v)$ denotes the dense 3D displacement of pixel $(u, v)$ from frame $s$ to frame $t$ in the world coordinate system, while $\mathcal{T}$ provides correspondences of sparse objects or points $\mathcal{O}_s$ to $\mathcal{O}_t$ within per-frame camera spaces.

While VGGT (Wang et al., 2025b) can predict geometry and 2D tracks jointly (thus indirectly inferring 3D motion via camera pose and depth), it still depends on an external tracker. Scaling sparse tracking to dense coverage would demand a huge number of queries, causing excessive memory and computational costs. To overcome these limitations, we propose a unified framework that jointly predicts geometry and motion in a single forward pass. Built on a geometry-aware backbone, it uses geometry heads and a motion head to simultaneously decode geometry and dense 3D motion, rather than the cascaded pipeline in Fig. 2.

Formally, given an input video or an image sequence $\mathcal{I} \in \mathbb{R}^{S \times 3 \times H \times W}$, our goal is to estimate both scene geometry (camera intrinsics $\mathcal{K}$, extrinsics $\mathcal{E}$, depth maps $\mathcal{D}$, and point maps $\mathcal{P}$) and dense 3D motion fields $\mathcal{V}$. Specifically:

$$\{\mathcal{K}, \mathcal{E}, \mathcal{D}, \mathcal{P}, \mathcal{V}\} = f_\theta(\mathcal{I}), \tag{3}$$

where $f_\theta$ denotes our unified network. For each source frame $s \in \{1, \ldots, S\}$ and target query frame $t \in \{1, \ldots, T\}$, the motion field is defined as:

$$\mathcal{V}_{s \to t}(u, v) = (x, y, z)_{s \to t}, \quad \forall (u, v) \in \{1, \ldots, H\} \times \{1, \ldots, W\}. \tag{4}$$

This formulation yields dense world-coordinate 3D motion for every pixel, enabling fine-grained modeling of dynamic interactions across the entire scene.

### 3.2 GEOMETRY-AWARE BACKBONE WITH MOTION HEAD

As illustrated in Figure 3, we present the overall architecture of our MotionVGGT, which comprises a geometry-aware backbone and a novel motion head. Given videos or image sequences captured within the same scene, our model not only reconstructs scene geometry, but also predicts dense, per-pixel 3D motion for all objects, which broadens the scope of visual geometry foundation models from 3D static scenes to 4D dynamic environments.

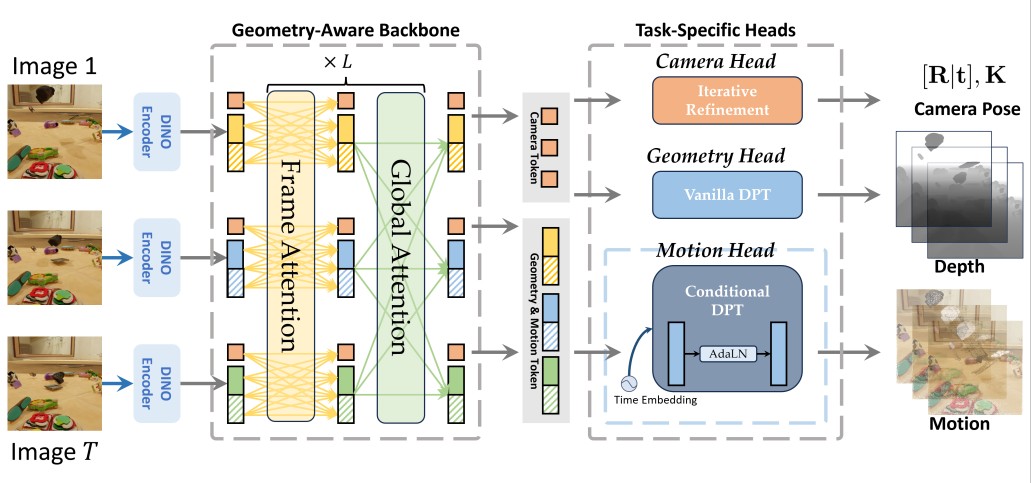

Figure 3: **Architecture of MotionVGGT.** It is composed of geometry-aware backbone and task-specific heads. The motion head takes the aggregated feature and time embedding as input.

**Geometry-Aware Backbone.** Given an input video or an image sequence $\mathcal{I} \in \mathbb{R}^{S \times 3 \times H \times W}$, we first patchfy each input image $I_s$ and extract their features $F_s \in \mathbb{R}^{N \times C}$ via a pretrained DINO encoder. For each frame, we append a learnable camera token $c_s \in \mathbb{R}^C$ and $K$ motion tokens $M_s \in \mathbb{R}^{K \times C}$ to the image tokens, then feed the sequence through a geometry-aware backbone that enables global cross-frame interactions and produces tokens enriched with 3D geometric cues. All these generated geometry tokens $G \in \mathbb{R}^{S \times (1+K+N) \times C}$ are then utilized for predicting dense 3D motion of objects in dynamic scenes. We adopt the pretrained VGGT as our geometry-aware backbone, as it is trained to predict core 3D attributes, including camera intrinsics and extrinsics, depth maps, point maps, and 3D point tracks, from one to hundreds of views in a single forward pass. This multi-task, multi-view supervision forces intermediate representations to encode epipolar geometry and cross-frame consistency, yielding features that are naturally aligned with 3D scene structure.

**Motion Head.** We reckon that the geometry tokens produced by a pretrained VGGT already encode rich motion information, as evidenced by the strong performance of VGGT features when coupled with CoTracker. However, reliance on an external tracker confines predictions to sparse 2D correspondences and cannot explicitly produce a dense 3D motion field. When scaled to dense tracking task, the number of query points grows dramatically, resulting in extremely high computational overhead. To simplify and generalize this architecture, we introduce a novel motion head that directly consumes the geometry-aware tokens and decodes them into dense, per-pixel 3D motion for all objects in a single forward pass. Specifically, we design a conditional Dense Prediction Transformer (DPT) head (Ranftl et al., 2021) that enhances geometry and temporal awareness during motion decoding. For each source frame $I_s$ in the input sequence, the per-frame motion tokens $M_s$, along with the query target time $t_q$, are injected into the image tokens $F_s$ through adaptive layer normalization (AdaLN) (Peebles & Xie, 2023) before applying DPT convolutions. As a result, for $S$ source frames and $T$ target query time, the motion head outputs a dense 3D motion field $\mathcal{V}$ of shape $[S, T, 3, H, W]$. This conditional decoding process can be formulated as:

$$\mathcal{V} = \text{DPTHead}\left(\text{AdaLN}(F, R, t_q)\right), \tag{5}$$

where $F = \{F_s\}_{s=1}^{S}$ denotes image tokens from source frames, $M = \{M_s\}_{s=1}^{S}$ denotes the corresponding motion tokens, and $t_q = \{t_1, t_2, \ldots, t_T\}$ is the set of query target times. This design eliminates reliance on external tracking modules while preserving motion-perception capability and extends visual geometry foundation models from static 3D perception to dynamic 4D understanding.

### 3.3 Multi-stage Training

We adopt a three-stage training strategy to exploit diverse supervision signals.

**Stage 1: Supervised pretraining with 2D tracking.** We start with datasets providing 2D tracking annotations. Given tracked correspondences between source and target frames, we reconstruct per-frame 3D point clouds using the pretrained VGGT backbone, then compute ground-truth 3D motion vectors. These serve as supervision for the motion head:

$$\mathcal{L}_{\text{motion}} = \frac{1}{|\Omega|} \sum_{i \in \Omega} \lambda_{\text{pt}} \|\hat{\mathbf{v}}_i - \mathbf{v}_i\|_1 + \frac{1}{|\Omega|^2} \sum_{(i,j) \in \Omega \times \Omega} \lambda_{\text{dist}} \|\hat{\mathbf{v}}_i \cdot \hat{\mathbf{v}}_j^\top - \mathbf{v}_i \cdot \mathbf{v}_j^\top\|_1. \tag{6}$$

where $\hat{\mathbf{v}}_i \in \mathbb{R}^3$ is the predicted 3D motion vector for point $i$, $\mathbf{v}_i \in \mathbb{R}^3$ is the corresponding ground-truth motion vector, $\Omega$ is the set of visible and matched points between two frames, $\lambda_{\text{pt}}$ and $\lambda_{\text{dist}}$ are balancing weights. To complement the direct $\ell_1$ loss , we follow (Lin et al., 2025) and introduce a distribution loss $\mathcal{L}_{\text{dist}}$ that encourages the predicted motion vectors to preserve relative pairwise distances within each frame, ensuring consistency and stability even under sparse supervision.

**Stage 2: Self-supervised learning with reprojection.** We further train on large-scale dynamic videos without explicit 2D tracks, using a reprojection-based consistency loss:

$$\mathcal{L}_{\text{reprojection}} = \frac{1}{|\Omega|} \sum_{i=1}^{\Omega} \|I_t(\pi(\mathbf{x}_i + \hat{\mathbf{v}}_i)) - I_s(p_i^s)\|_2, \tag{7}$$

$$\mathcal{L}_{\text{grad}} = \frac{1}{N} \sum_{p \in N} \left( \left\|\nabla_x \hat{\mathbf{V}}(p)\right\|_1 + \left\|\nabla_y \hat{\mathbf{V}}(p)\right\|_1 \right), \tag{8}$$

$$\mathcal{L}_{\text{total}} = \mathcal{L}_{\text{reprojection}} + \lambda_{\text{grad}} \mathcal{L}_{\text{grad}}, \tag{9}$$

where $\mathbf{x}_i \in \mathbb{R}^3$ is the 3D coordinate of the $i$-th source-frame point, $\hat{\mathbf{v}}_i \in \mathbb{R}^3$ is the predicted 3D motion vector from source to target, $\pi(\cdot)$ projects 3D world coordinates to 2D using target intrinsics/extrinsics, $I_t(\cdot)$ and $I_s(\cdot)$ return RGB values from the target and source frames, respectively, $p_i^s$ is the 2D source location corresponding to $\mathbf{x}_i$, $\Omega$ is the number of valid (visible and projectable) points, $N$ is the total number of per-frame pixels and $p \in N$ indexes a pixel, $\hat{\mathbf{V}}(p) \in \mathbb{R}^3$ denotes the predicted 3D motion vector at pixel $p$, $\nabla_x, \nabla_y$ are first-order forward differences, $\lambda_{\text{grad}}$ is a weighting factor; and $\mathcal{L}_{\text{grad}}$ serves as a regularizer enforcing local spatial coherence of the motion field (a physical prior that encourages neighboring pixels to have consistent motion).

**Stage 3: Joint finetuning of geometry and motion.** Finally, we finetune all heads (camera, depth, point map, motion) on all dynamic datasets. Geometry losses follow VGGT (Wang et al., 2025b), while the motion loss combines the supervised objective from Stage 1 (when annotations exist) and the reprojection loss from Stage 2. This joint optimization allows geometry and motion information to be mutually reinforced, improving performance across multiple tasks under dynamic scenarios.

## 4 EXPERIMENTS

### 4.1 IMPLEMENTATION DETAILS

The geometry-aware backbone of MotionVGGT is built upon a pretrained VGGT model consisting of 24 layers of global and frame attention modules. During training, the backbone is kept frozen while the Motion Head is trained using the AdamW optimizer for 10 epochs in Stage I, II, and III, respectively. The total number of trainable parameters in MotionVGGT is approximately 200 million. Following VGGT, we adopt a cosine learning rate scheduler with a peak learning rate of 0.0002. Each training batch consists of 8 consecutive frames sampled from diverse training scenes. The input RGB images are resized to a maximum dimension of 518 pixels. Training is conducted on 4 NVIDIA A800 GPUs over 4 days. To optimize GPU memory usage and computational speed, we employ bfloat16 precision and gradient checkpointing.

### 4.2 TRAINING DATASETS

MotionVGGT is trained on a selected multi-domain dataset collection consisting of 8 dynamic datasets. For training stage 1, we use Kubric (Greff et al., 2022), DynamicReplica (Karaev et al., 2023), and PointOdessey (Zheng et al., 2023), while stage 2 appends Waymo (Sun et al., 2020),

Table 1: **Training Dataset Description.** ✔ and ✗ indicate the actual usage in our training process.

| Dataset | Composition | | | | Utilization | | |
|---|---|---|---|---|---|---|---|
| | RGB | Depth | Track | Pose | Stage I | Stage II | Stage III |
| PointOdessey (Zheng et al., 2023) | ✔ | ✔ | ✔ | ✔ | ✔ | ✔ | ✔ |
| DynamicReplica (Karaev et al., 2023) | ✔ | ✔ | ✔ | ✔ | ✔ | ✔ | ✔ |
| Kubric (Greff et al., 2022) | ✔ | ✗ | ✔ | ✗ | ✔ | ✔ | ✔ |
| Waymo (Sun et al., 2020) | ✔ | ✔ | ✗ | ✔ | ✗ | ✔ | ✔ |
| VKitti2 (Cabon et al., 2020) | ✔ | ✔ | ✗ | ✔ | ✗ | ✔ | ✔ |
| TUM-Dynamics (Sturm et al., 2012) | ✔ | ✗ | ✗ | ✗ | ✗ | ✔ | ✔ |
| Bonn (Palazzolo et al., 2019) | ✔ | ✗ | ✗ | ✗ | ✗ | ✔ | ✔ |
| Sintel (Butler et al., 2012) | ✔ | ✗ | ✗ | ✗ | ✗ | ✔ | ✔ |

Table 2: **Video depth evaluation** on Sintel, Bonn, and KITTI datasets.

| Method | Sintel | | Bonn | | KITTI | |
|---|---|---|---|---|---|---|
| | AbsRel ($\downarrow$) | $\delta_{1.25}$ ($\uparrow$) | AbsRel ($\downarrow$) | $\delta_{1.25}$ ($\uparrow$) | AbsRel ($\downarrow$) | $\delta_{1.25}$ ($\uparrow$) |
| DUSt3R-GA (Wang et al., 2024) | 0.656 | 45.2 | 0.155 | 83.3 | 0.144 | 81.3 |
| MASt3R-GA (Leroy et al., 2024b) | 0.641 | 43.9 | 0.252 | 70.1 | 0.183 | 74.5 |
| MonST3R-GA (Zhang et al., 2024) | 0.378 | 55.8 | 0.067 | 96.3 | 0.168 | 74.4 |
| VGGT (Wang et al., 2025b) | 0.298 | 68.1 | 0.057 | 96.8 | **0.061** | 97.0 |
| Spann3R (Wang & Agapito, 2024) | 0.622 | 42.6 | 0.144 | 81.3 | 0.198 | 73.7 |
| CUT3R (Wang et al., 2025c) | 0.421 | 47.9 | 0.078 | 93.7 | 0.118 | 88.1 |
| Point3R (Wu et al., 2025) | 0.452 | 48.9 | 0.060 | 96.0 | 0.136 | 84.2 |
| StreamVGGT (Zhuo et al., 2025) | 0.323 | 65.7 | 0.059 | 97.2 | 0.173 | 72.1 |
| MotionVGGT (Ours) | **0.294** | **69.3** | **0.056** | **97.4** | 0.065 | **97.5** |

VKitti2 (Cabon et al., 2020), Bonn (Palazzolo et al., 2019), TUM-Dynamics (Sturm et al., 2012), and Sintel (Butler et al., 2012). These datasets cover a wide range of visual domains, including both indoor and outdoor dynamic scenes, as well as varying temporal scales. This diversity ensures that the model generalizes well across different levels of geometric complexity and varying viewpoints. Since the additional datasets introduced in training stage 2 lack tracking annotations, we apply a reprojection loss for these data.

## 4.3 PERFORMANCE EVALUATION

We evaluate our proposed **MotionVGGT** framework across multiple downstream tasks in motion perception, including 3D motion prediction, video depth estimation, and 4D reconstruction in real-world dynamic environments. We compare against state-of-the-art methods, ranging from optical flow models to recent video foundation models and 3D reconstruction systems. All metrics are reported using standard benchmarks and evaluation protocols.

### 4.3.1 VIDEO DEPTH EVALUATION

We evaluated monocular video depth estimation on three diverse benchmarks: KITTI (Geiger et al., 2013), TUM Dynamics (Sturm et al., 2012), Bonn RGB-D (Palazzolo et al., 2019), Sintel (Butler et al., 2012), and an averaged metric across datasets. We report **AbsRel** (absolute relative error) and $\delta_{1.25}$, following standard practice.

Table 2 summarizes the results. Following the protocol of CUT3R, our MotionVGGT excels in both accuracy and consistency across domains. It achieves the lowest AbsRel and highest $\delta_{1.25}$ on average, indicating that joint motion-depth learning within a unified transformer backbone enhances depth reasoning in dynamic scenes.

Table 3: **Quantitative dynamic scene reconstruction results on TUM-dynamics.**

| Method | Type | Acc ($\downarrow$) Mean | Acc ($\downarrow$) Med | Comp ($\downarrow$) Mean | Comp ($\downarrow$) Med | NC ($\uparrow$) Mean | NC ($\uparrow$) Med |
|---|---|---|---|---|---|---|---|
| VGGT (Wang et al., 2025b) | Dense-view | 0.050 | 0.008 | 0.055 | 0.017 | **0.622** | **0.695** |
| CUT3R (Wang et al., 2025d) | Streaming | 0.105 | 0.012 | 0.060 | **0.007** | 0.582 | 0.624 |
| StreamVGGT (Zhuo et al., 2025) | Streaming | 0.085 | 0.011 | 0.058 | **0.007** | 0.617 | 0.690 |
| MotionVGGT (Ours) | Dense-view | **0.047** | **0.007** | **0.035** | 0.011 | 0.582 | 0.626 |

Table 4: **Optical flow results on CVO dataset.**

| Method | CVO Clean EPE $\downarrow$ (*all/vis/occ*) | CVO Clean IoU $\uparrow$ | CVO Final EPE $\downarrow$ (*all/vis/occ*) | CVO Final IoU $\uparrow$ |
|---|---|---|---|---|
| RAFT (Teed & Deng, 2020) | 2.48/1.40/7.42 | 57.6 | 2.63/1.57/7.50 | 56.7 |
| MFT (Neoral et al., 2024) | 2.91/1.39/9.93 | 19.4 | 3.16/1.56/10.3 | 19.5 |
| TAPIR (Doersch et al., 2023) | 3.80/1.49/14.7 | 73.5 | 4.19/1.86/15.3 | 72.4 |
| CoTracker (Karaev et al., 2024) | 1.51/0.88/4.57 | 75.5 | 1.52/0.93/4.38 | 75.3 |
| DOT (Le Moing et al., 2024) | 1.29/0.72/4.03 | 80.4 | 1.34/0.80/3.99 | 80.4 |
| SceneTracker (Wang et al., 2025a) | 4.40/3.44/9.47 | –.– | 4.61/3.70/9.62 | –.– |
| SpatialTracker (Xiao et al., 2024) | 1.84/1.32/4.72 | 68.5 | 1.88/1.37/4.68 | 68.1 |
| DOT-3D (Le Moing et al., 2024) | 1.33/0.75/4.16 | 79.0 | 1.38/0.83/4.10 | 78.8 |
| DELTA-2D (Ngo et al., 2025) | 0.89/0.46/2.96 | 78.3 | 0.97/0.55/2.96 | 77.7 |
| DELTA-3D (Ngo et al., 2025) | 0.94/0.51/2.97 | 78.7 | 1.03/0.61/3.03 | 78.3 |
| MotionVGGT (Ours) | 9.14/6.21/17.18 | 40.0 | 9.19/5.99/17.53 | 40.0 |

### 4.3.2 RECONSTRUCTION IN DYNAMIC SCENARIOS

To evaluate 3D reconstruction quality in dynamic environments, we conducted experiments on the TUM-Dynamics dataset (Sturm et al., 2012), which contains real-world scenes with moving objects and articulated motions. We assess both geometric accuracy and completeness using point-to-mesh distance metrics: **Accuracy** (Acc), **Completeness** (Comp), and **Normal Consistency** (NC).

As shown in Table 3, our method achieves satisfying performance in reconstructing dynamic scenes from monocular video. This demonstrates that integrating motion perception directly into the reconstruction pipeline leads to more coherent and stable geometry reconstruction.

### 4.3.3 2D POINT TRACKING

We assessed motion prediction accuracy on the CVO dataset (Wu et al., 2023; Le Moing et al., 2024), which provides dense ground-truth flow annotations under both clean and complex motion conditions. Following prior work, we report two primary metrics: **End Point Error (EPE)** in pixels ($\downarrow$), broken down into overall (*all*), visible (*vis*), and occluded (*occ*) regions; and **Intersection over Union (IoU)** for motion segmentation ($\uparrow$), which evaluates the consistency of predicted motion clusters.

As shown in Table 4, MotionVGGT demonstrates generalization in handling object motions. It leverages global and frame-wise attention with explicit motion modeling, enabling robust tracking.

### 4.3.4 3D MOTION PREDICTION

We evaluate dense 3D motion prediction on the Kubric (Greff et al., 2022) test split generated by DELTA (Ngo et al., 2025). The dataset has 143 synthetic videos with a length of 24 frames. Following prior work, we report **Average Jaccard (AJ)** ($\uparrow$), which measures both occlusion and position accuracy, $\mathbf{APD_{3D}}$ ($\uparrow$), which measures the percentage of points within the error thresholds, and **Occlusion Accuracy (OA)** ($\uparrow$), which measures the accuracy of visibility/occlusion prediction.

Table 5: **Dense 3D tracking results on the Kubric3D dataset.**

| Methods | Kubric-3D (24 frames) | | |
|---|---|---|---|
| | AJ↑ | $APD_{3D}$ ↑ | OA↑ |
| SpatialTracker | 42.7 | 51.6 | 96.5 |
| SceneTracker | - | 65.5 | - |
| DOT-3D | 72.3 | 77.5 | 88.7 |
| DELTA | **81.4** | **88.6** | **96.6** |
| Ours | 13.9 | 24.6 | 78.6 |

Table 6: **Impact of Reprojection-Based Self-Supervision on Motion Perception.**

| Methods | Kubric-3D (24 frames) | | |
|---|---|---|---|
| | AJ↑ | $APD_{3D}$ ↑ | OA↑ |
| Ours (I) | 13.8 | 24.5 | 78.6 |
| Ours (I+II) | **13.9** | **24.6** | **78.6** |

Table 7: **Stage-III Improves Geometry in Dynamic Scenes (TUM-Dynamics).**

| Method | Acc (↓) | | Comp (↓) | | NC (↑) | |
|---|---|---|---|---|---|---|
| | Mean | Med | Mean | Med | Mean | Med |
| Ours (I+II) | 0.050 | 0.008 | 0.055 | 0.017 | **0.622** | **0.695** |
| Ours (I+II+III) | **0.047** | **0.007** | **0.035** | **0.011** | 0.582 | 0.626 |

Table 8: **Effect of Motion-Token Conditioning on 3D Motion Perception.**

| Methods | Kubric-3D (24 frames) | | |
|---|---|---|---|
| | AJ↑ | $APD_{3D}$ ↑ | OA↑ |
| ours (W/O Motion Tokens) | 13.4 | 23.8 | 78.6 |
| Ours (W/ Motion tokens) | **13.8** | **24.5** | **78.6** |

## 4.4 ABLATION STUDY

**Multi-stage training strategy.** To validate the effectiveness of multi-stage training strategy, we conducted two additional studies. First, we examined the contribution of the reprojection–based second training stage to 3D motion perception. We compared our model which was trained only with the first stage against a counterpart further optimized with the second stage and evaluated both on Kubric3D (Greff et al., 2022). As summarized in Table 6, the two-stage model achieves higher accuracy in predicting 3D motion, confirming the utility of the reprojection loss. By contrast, incorporating the second stage with self-supervised training on driving-scene datasets enables reliable motion prediction in this domain, underscoring the necessity of reprojection-based supervision. Second, we evaluated the role of the third training stage, which integrates motion information to guide geometry in dynamic scene reconstruction. As reported in Table 7, the model trained through the third stage delivers consistently higher reconstruction fidelity in dynamic settings than the two-stage baseline, demonstrating that leveraging motion information further improves geometry perception.

**Motion tokens as conditions.** To assess the efficacy of injecting motion tokens into our motion head, we performed a controlled ablation on the Kubric3D (Greff et al., 2022) benchmark. We compared the full model, where the motion tokens condition our motion head, against an otherwise identical variant in which this conditioning is removed. As reported in Table 8, the conditioned model achieves consistently higher 3D motion accuracy , demonstrating that motion tokens can provide informative, discriminative cues about scene dynamics.

## 5 CONCLUSION

In this work, we proposed **MotionVGGT**, an end-to-end framework for unified dense 3d geometry and motion perception. By injecting motion tokens into the conditional DPT head, features from the geometry-aware backbone are decoded into dense motion fields in world coordinates, obviating the need for cascaded pipelines and mitigating the sparse query design and first frame dependency that characterize traditional tracking methods. Extensive experiments show that dense motion perception improves reconstruction quality in dynamic scenes, underscoring the importance of motion information for geometry understanding. These properties make MotionVGGT well-suited for autonomous driving, embodied robotics, and AR/VR applications, where robust dynamic scene reconstruction is essential.

**Limitations.** The model is trained on fixed-length sequences, leading to significant performance degradation when applied to longer, dynamic sequences during inference. Besides, the model exhibits insufficient capability in accurately tracking objects undergoing rapid movement. These remain to be investigated for us in the future. Meanwhile, due to the scarcity of samples, our method underperforms in driving scenes.

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
