# OpenReview forum: "End-to-End Unified Dense 3D Geometry and Motion Perception"
_ICLR.cc/2026/Conference — ICLR 2026 Conference Withdrawn Submission_

### Official Review · Reviewer_nuhf · 2025-10-26

**Soundness:** 1
**Presentation:** 3
**Contribution:** 1
**Rating:** 2
**Confidence:** 5

**Summary:**

The paper proposes MotionVGGT, a VGGT-based architecture that jointly predicts dense 3D geometry (cameras, depth, point maps) and dense 3D motion fields in world coordinates in a single forward pass. It augments a geometry-aware backbone with motion tokens and a conditional dense prediction head to decode per-pixel 3D motion; training proceeds in three stages (supervised with 2D tracks, self-supervised via reprojection, and joint finetuning). Experiments cover video depth on Sintel/Bonn/KITTI, dynamic reconstruction on TUM-Dynamics, optical flow on CVO, and dense 3D tracking on Kubric.

**Strengths:**

* **Joint geometry–motion formulation**. Unifying camera/depth/point-map prediction with dense 3D motion in a single forward pass targets an important and challenging problem with clear real-world relevance.

* **Clear architecture**. The addition of motion tokens and a conditional head (AdaLN-conditioned DPT) is straightforward and easy to follow, making the design transparent.

**Weaknesses:**

* **Limited technical novelty**. The pipeline closely mirrors St4rtrack (ICCV’25)—a motion head on top of an existing reconstruction model—here simply transplanted onto the stronger VGGT backbone with largely similar design, training, and objectives. The added motion tokens and AdaLN-DPT appear ineffective based on ablations (see below), offering little architectural innovation.

* **Evaluation results**.
    1. Data leakage: Foundational geometry models are typically evaluated zero-shot on Sintel and TUM-Dynamics, yet these datasets appear in the training phase here; consequently, results in Table 2 (first column), Table 3, and Table 7 are not comparable to prior zero-shot reports.

     2. Weak motion quality: On Kubric3D and CVO, the method lags substantially behind baselines, indicating limited ability to capture dynamic motion.

     3. Ablations with minimal effect: Ablations for the motion design (e.g., with/without motion tokens, reprojection loss) yield near-identical results, suggesting the proposed components do not support the gains.

* **Generalizability claims overstated**: Given the weak performance on synthetic motion benchmarks, it is difficult to accept the broad claim of “strong generalization across diverse geometry perception tasks”

**Questions:**

Given the weak results on both 2D/3D motion benchmarks, have you explored that first trains the network to predict dense 2D motion (e.g., optical flow over short time), then lifts to 3D motion, similar to [1]? If so, please report settings and whether such pretraining improves performance on Kubric3D/CVO. If not, could you comment on why this path was not pursued and whether a zero-shot 2D motion pretraining (e.g., on Sintel/FlyingThings3D or large-scale in-the-wild flow, as in [1]) might help your 3D motion quality and generalization?


**Conclusion**

Overall, the paper reads as an incremental extension of St4rtrack, with the main change being a swap to the stronger VGGT backbone. The evaluation is insufficient: it omits comparisons to state-of-the-art dense 3D tracking, relies on datasets overlapping with training, and lacks diverse real-world tests. Thus I recommend reject.

[1] Liang, Yiqing et al., “Zero-Shot Monocular Scene Flow Estimation in the Wild,” CVPR 2025, pp. 21031–21044.

---

### Official Review · Reviewer_fCRw · 2025-10-31

**Soundness:** 1
**Presentation:** 1
**Contribution:** 2
**Rating:** 2
**Confidence:** 4

**Summary:**

The paper presents MotionVGGT, an end-to-end transformer that jointly predicts dense 3D geometry, camera pose, and per-pixel 3D motion in a single forward pass, eliminating the need for external trackers.

Each video frame is encoded into geometry, camera, and motion tokens, which interact via interleaved frame-attention and global-attention layers. Dedicated heads then decode point-cloud maps, camera poses, and motion fields. A conditional dense-prediction head leverages motion tokens to modulate geometry tokens, coupling geometry and motion estimation.

By unifying these two aspects, MotionVGGT brings motion awareness to visual-geometry foundation models.

**Strengths:**

By discarding the prevailing reconstruct-then-track paradigm and its iterative trackers, the paper offers a unified formulation of dynamic-scene perception.

The method needs no optical flow, external tracker, or extra geometric cues, yet produces image-wide dense motion.

By equipping visual-geometry foundation models with motion awareness, MotionVGGT is a simple path from static to dynamic scene understanding.

**Weaknesses:**

The interleaved frame-attention and global-attention design is inherited almost verbatim from VGGT; listing it as a new contribution is misleading. Please clarify what is genuinely new (e.g., the motion) and explicitly separate it from prior work.

Fig. 1 is the only visualization, and it shows barely perceptible motion against an almost static background. More diverse, high-motion scenes (e.g., human actions, street traffic) are needed to substantiate claims.

Fig. 2 mostly reiterates ideas already covered in TAPIP3D, it does not illustrate MotionVGGT’s unique elements. Consider replacing it with a more focused schematic.

**Questions:**

Lack of the visualization of 4D results. Images and video results are crucial for evaluating 4D reconstruction. At present, none are provided, which makes the submission feel premature.

The paper does not discuss or compare against several closely related and contemporaneous methods:

    St4RTrack: Simultaneous 4D Reconstruction and Tracking in the World
    TAPIP3D: Tracking Any Point in Persistent 3D Geometry
    Easi3R: Estimating Disentangled Motion from DUSt3R Without Training
    MoVieS: Motion-Aware 4D Dynamic View Synthesis in One Second
    POMATO: Marrying Pointmap Matching with Temporal Motions for Dynamic 3D Reconstruction
    SpatialTrackerV2: 3D Point Tracking Made Easy
    TTT3R: 3D Reconstruction as Test-Time Training
    Trace Anything Representing Any Video in 4D via Trajectory Fields
    Depth Anything 3: Recovering the Visual Space from Any Views

Could the authors clarify how MotionVGGT differs from, or improves upon, these approaches and, if possible, include quantitative/qualitative comparisons?

---

### Official Review · Reviewer_ZN6p · 2025-10-31

**Soundness:** 2
**Presentation:** 3
**Contribution:** 2
**Rating:** 4
**Confidence:** 4

**Summary:**

The paper proposes MotionVGGT, extending a pretrained geometry-aware transformer (VGGT) with motion tokens and a conditional DPT head to jointly predict camera parameters, depth/point maps, and dense world-coordinate 3D motion in a single forward pass. Trained in three stages (supervised 2D tracking, self-supervised reprojection, and joint fine-tuning), it offers a practical, unified approach to dynamic scene understanding.

**Strengths:**

The paper addresses an important problem by extending geometry foundation models into a unified, motion-aware framework. The design is elegant and practical, enabling dense motion prediction in a single forward pass without external trackers. The multi-source, multi-stage training scheme effectively combines supervised and self-supervised learning across diverse datasets.

**Weaknesses:**

1) The core concern lies in the lack of methodological novelty. The proposed approach mainly adds a DPT head for motion prediction and conditional modulation on top of VGGT. This design is rather incremental and has been widely seen in recent vision transformer works.

2) The experimental results conflict with the paper’s main claims. In Table 4 and Table 5, MotionVGGT performs significantly worse than its backbone (and other baselines), especially on motion-related benchmarks, which are supposed to be its core strength. This weakens the claimed advantage of "unified motion perception." The paper repeatedly claims "strong generalization," but the numbers clearly show otherwise.

3) Although the authors briefly acknowledge performance degradation in the Limitations section, they do not analyze why it happens. The experimental discussion is superficial, and there are no qualitative visualizations to substantiate the claims.

4) The writing is clear but contains noticeable redundancy, e.g., the Introduction and Method sections repeatedly emphasize "unified" and "dense" without adding substantive insight.

**Questions:**

1) Section 4.1 states that the VGGT backbone is frozen and only the motion head is trained, while Section 3.3 mentions that Stage III involves joint fine-tuning of geometry and motion. This inconsistency should be clarified.

2) Some key implementation details are missing, for example, how many motion tokens are used, and whether their number is fixed or adaptive across frames.

---

### Official Review · Reviewer_evDg · 2025-11-01

**Soundness:** 2
**Presentation:** 2
**Contribution:** 1
**Rating:** 2
**Confidence:** 5

**Summary:**

The paper proposes a joint 3D geometry and motion perception method.
Given a pretrained geometry foundation model, VGGT, the method attaches a motion head to predict 3D scene flow of each point.
The paper proposes multi-stage training techniques and achieves reasonable accuracy on both geometry and motion benchmarks.

---

There are lots of concerns found in the paper: lack of novelty (incremental architectural improvement), no comprehensive ablation study (unclear if the proposed idea is effective), insufficient/invalid experiment setup (no qualitative results, test set is used during training). Thus the recommendation of the paper is **2. Reject**.

**Strengths:**

* **Good clarity**

  The paper is written clearly. All the details are easy to follow.

* **Good motivation for joint 3D geometry and 3D motion**

  The introduction part nicely motivate the necessity of joint 3D geometry and 3D motion estimation, by comparing with previous methods of multi-stage pipelines (eg., estimation of geometry first, followed by motion estimation).

**Weaknesses:**

* **Novelty concern**

  The architecture is a combination of VGGT and DynaDUSt3R [1]. The motion head from DynaDUSt3R is adopted in the VGGT architecture. It's difficult to find any new innovations for this problem of joint 3D geometry and 3D motion estimation. The newly proposed part is the multi-stage training in section 3.3 only, but the effectiveness of the multi-stage training is also questionable (continued in the other bullet points).

  [1] Stereo4D: Learning How Things Move in 3D from Internet Stereo Videos

* **Multi-stage training and ablation study**

  It is unclear if the multi-stage training is actually effective. In Table 6, the improvement is very marginal, 0.1 point, or the same. It's not so sure if this level of improvement can be considered as "effective". For better thoroughness, it is recommended to share the full ablation study. It can be:
  * Stage1 only
  * Stage2 only
  * Stage1 + Stage2
  * Stage1 + Stage3
  * Stage2 + Stage3
  * Stage1 + Stage2 + Stage3

  and evaluate on both geometry and motion estimation tasks. Without it, it's hard to believe if this multi-stage training is actually needed. This also results in the lack of novelty concerns.


* **Poor motion accuracy**

  In Table 4 and 5, the 3D tracking accuracy heavily underperforms, comparing with other methods. 3D tracking of the method seems not properly working.

* **TUM-dynamics dataset**

  Table 3 reports the accuracy of the method on TUM-dynamics dataset. However, the dataset is actually included in the training sets (Table 1), which questions the validity of the results.


* **No qualitative results**

  There is no qualitative comparison attached in the paper. It is difficult to understand how the method behave.

**Questions:**

* **In Figure 1**, what's the difference between images in the 2nd row and that in the 3rd row?

* **In Equation 2**, the 3D tracking method can also recover the camera pose via post processing because it has the 3D correspondence. I am not so sure if this is truly a critical limitation of the 3D Tracking approach.

* **In Equation 7**, when projecting points into image coordinate, it's likely that multiple points can be projected into a single pixel coordinate. In this case, how does the method handle depth ordering?

---

### Note · Authors · 2025-11-20

I have read and agree with the venue's withdrawal policy on behalf of myself and my co-authors.